# Non-guided, mobile, CBT-I-based sleep intervention in War-torn Ukraine: A feasibility study

Anton Kurapov[1,2]*, Jens Blechert[3], Alexandra Hinterberger[1], Pavlos Topalidis[1], Manuel Schabus[1]

1 Laboratory for Sleep, Cognition and Consciousness Research, Paris Lodron University of Salzburg, Austria, 2 Department of Experimental and Applied Psychology, Faculty of Psychology, Taras Shevchenko National University of Kyiv, Ukraine, 3 Division of Clinical Psychology and Health Psychology, Paris Lodron University of Salzburg, Austria

* anton.kurapov@plus.ac.at

## Abstract

### Background

War conditions can severely impact sleep and mental health at the population level, especially in the conflicts of such tremendous scale as in Ukraine. The aim of this research was to study whether a mobile, unguided Cognitive Behavioral Therapy-based Intervention for sleep problems, Sleep2, is feasible, acceptable, and potentially able to reduce mental health/sleep problems symptoms.

### Methods

A single-arm, open-label, uncontrolled pre-post evaluation study was conducted with 487 registered participants: 283 started, 160 (56.55%) finished, out of which 95 completed without an ambulatory heart rate (HR) sensor and 65 with. Assessments were conducted through online questionnaires and objective measurements via HR sensors. Besides feasibility and acceptability, outcome measures included symptoms in several mental health domains alongside self-reported and objectively reported sleep parameter.

### Results

Engagement with the Sleep2 app was high, achieving an 80.72% compliance rate, alongside high levels of feasibility and acceptance. Participants reported significant pre-post reductions in the severity of symptoms, with sleep problems decreasing by 22.60% (Cohen's $d = 0.53$), insomnia by 35.08% ($d = 0.69$), fear of sleep by 32.43% ($d = 0.25$), anxiety by 27.72% ($d = 0.48$), depression by 28.67% ($d = 0.52$), PTSD by 32.41% ($d = 0.51$), somatic symptoms by 24.52% ($d = 0.51$), and perceived stress by

**Data availability statement:** Data is publicly available via OSF repository: https://osf.io/5n3cb/.

**Funding:** This research was funded in whole or in part by the Austrian Science Fund (FWF) [10.55776/W1233].

**Competing interests:** Authors declare no conflict of interests, except for Manuel Schabus, who is Co-Founder and CSO of NUKKUAA®, which does not alter our adherence to PLOS ONE policies on sharing data and materials.

17.90% ($d = 0.39$). Objective sleep measurements showed a significant reduction in sleep onset latency only.

## Conclusion

The 'Sleep2Ukraine' program demonstrated high feasibility and acceptance, with significant improvements in subjective sleep and mental health measures among participants. However, given the study's uncontrolled design and reliance on self-selected participants, these findings should be considered preliminary. Randomized controlled trials are needed to establish efficacy. Nonetheless, the results highlight the potential of culturally adapted, scalable, mobile-based CBT-I interventions to address sleep and mental health needs in war-affected populations.

## Introduction

War can be considered a specific form of crisis that inevitably impacts various aspects of life [1], including mental health [2] in both military personnel and civilians. Besides typical trauma-related disorders such as anxiety, depression, PTSD, civil victims of war were shown to develop long-term sleeping problems that may last for years even after the conflict termination [3,4]. Thus, as for sleeping problems, war has a significant impact on health via (at least) two pathways: through direct causation of trauma related disorders, as well as indirectly by disturbing resilience factors such as healthy and restorative sleep. It is also well known that sleep issues reinforce mental health complaints [5], with unrestorative sleep being the key maintenance factor for anxiety, depression, and PTSD [6]. In turn, improvement of sleep quality usually improves mental health symptoms in general [7,8].

As of July 2024, Ukraine has been at war for more than two years. Respective recent studies showed significant deterioration of mental health: from the very beginning of the war [9,10] and throughout its progression [11,12]. A specific problem seems to be the constant and unpredictable air attacks on the entire territory of Ukraine [13,14]. Being mostly carried out during nighttime to complicate air defense, these attacks disturb sleep directly through loud raid alarm systems and likely contribute to fear of sleep through a build-up of anxious apprehension and anticipation of further disturbances and the need to rush for shelter quickly or search for a room without windows at home. Therefore, the need for easily accessible and scalable sleep interventions and coaching for Ukrainians is evident and of a high need.

One of the most efficient sleep interventions is classical, face-to-face, cognitive behavioral therapy for insomnia (CBT-I) [15]. European [16] and US [17] clinical guidelines recommend CBT-I as the first line of treatment. It is a non-pharmacological, well-established intervention, teaching relaxation techniques, sleep restriction, stimulus control, psychoeducation, and cognitive strategies [16]. Recent meta-analyses show overall high efficiency of such non-pharmacological, conventional, face-to-face interventions [7,18]. Digital CBT-I (dCBT-I) represents the online implementations of these successful programs, with the advantage of being ubiquitously available and cost

effective [19]. Also, dCBT-I has been studied and has been shown to generate mostly comparable effects to its face-to-face counterpart [19,20]. In traumatized populations it showed improvements not only on sleep parameters [21,22], but also on other mental health indices [18,23]. As such, dCBT-I has also been successfully used in prevention [24].

As indicated above, one of the major advantages of mobile-based CBT interventions in Ukraine is their minimal logistical and time demands, as they do not require travel, specialized equipment, or intensive time commitment – features that are critical in war settings where resources and mobility are usually limited. War conditions make accessing specialized treatment centers difficult in some areas. Mental health resources are limited and focused on the most acute, physical condition, and the armed forces. Mobile interventions provide access to anyone with a smartphone and internet connection, thereby increasing accessibility, feasibility, and acceptance. However, while many scientifically proven interventions are currently available in English, there are very few that are both scientifically validated and freely available in Ukrainian. This lack of accessible resources in the local language may pose a significant barrier for non-English speakers, highlighting the importance of offering treatments in Ukrainian to enhance feasibility, acceptance, and overall effectiveness. To our knowledge, 'Sleep2Ukraine' is the only scalable mobile sleep intervention delivered to Ukrainians that specifically targets sleep problems and comes in Ukrainian language. The novel, evidence-based smartphone application Sleep2 (formerly called NUKKUAA®) comprises an unguided, fully automated CBT-I-based sleep treatment program. Sleep2 was initially developed in German for a general adult population but was later adapted for Ukrainian users to ensure cultural and linguistic relevance. One particular aspect of the application is the heart rate-based sleep scoring: using chest-belt-derived heart rate data and server-based matching algorithms, sleep stages can be inferred with near to human-expert classification accuracies [25,26]. In the morning, objective sleep architecture data are fed back to users, along with norm data comparisons and respective tailored strategies for sleep improvement. Yet, continuous sensor wearing can be bothersome, uncomfortable, and, generally, unguided interventions are known to yield only limited and short-term compliance [27].

To our knowledge, no studies to date have examined CBT-I specifically in war-affected populations, whether delivered digitally or face-to-face, particularly in regions where the conflict is ongoing or has not yet been resolved. Thus, in preparation for a formal randomized controlled efficacy trial, the present research aimed to assess the uptake, feasibility, acceptance, and usability of the Ukrainian version of Sleep2, provided free of charge to all Ukrainian users. The study was preregistered (see the link for details on hypotheses: https://osf.io/2gh93). The duration had to be extended from 4 to 6 weeks due to participants' non-daily use of the program and the need to complete most modules before post-assessment. Uptake was of a particular relevance as roll out in a population under constant exposure to war can be challenging, particularly in connection with ambulatory biofeedback devices such as the heart rate sensor. Thus, we tested our preregistered hypotheses expecting high levels of engagement, feasibility, and acceptance. Uptake was defined as the proportion of registered users who completed the intervention. Feasibility was measured through participant engagement, including app usage and completion of required exercises and diary entries. Acceptance was evaluated based on participant satisfaction scores regarding the app's usability, relevance, and subjective effectiveness. We expected self-selection to result in participants with heightened levels of sleep problems, our target population, and some mental health symptoms. Further, we predicted improvement of subjective sleep quality and objective sleep parameters such as sleep onset latency, number of awakenings, and sleep efficiency. We also collected qualitative data on app use. Lastly, given the relationship between sleep and mental health, as well as the potential generalization of CBT techniques, we predicted improvements in mental health symptoms (anxiety, depression, post-traumatic stress disorder, somatic symptoms).

## Methods

### Participants

This study targeted the Ukrainian population residing within Ukraine. Participants were eligible if they were aged between 18 and 75 years, fluent in Ukrainian, and owned an Android or iOS smartphone to access the Sleep2Ukraine mobile app. No additional exclusion criteria were applied. Recruitment was conducted through various channels, including

advertisements distributed by the Ukrainian Psychological Association, National Psychological Association of Ukraine, university networks, rehabilitation centers, non-profit organizations, and social media platforms such as Telegram and Facebook, aiming to include individuals from diverse demographic and geographic backgrounds across Ukraine. Participants self-enrolled in the study and did not receive financial compensation but were granted unrestricted access to the Sleep2Ukraine program. To ensure privacy, all data were collected anonymously as per the study design (see below) and for compliance with the General Data Protection Regulation. Recruitment and registration for the study began on October 1, 2023, and ended on November 1, 2023. During registration, participants could voluntarily choose whether they want to receive the heart rate (HR) sensor free of charge, but with the necessity to return it after 6 weeks at the latest. To maintain anonymity, the sensor was shipped to participant through a third-party service provider [The sensors were shipped to a responsible person in Ukraine, who then contacted participants to gather the necessary details for delivery. The delivery was carried out using the courier service "Nova Poshta" (Eng. "New Post"). Participants were given the option to either collect the belts at the post office or have them delivered directly to their address. As such, we did not know the identities of our participants and could not match their data with their personalities, as study emails were used to log into the app, which did not disclose any personal details, ensuring complete anonymity.] . The study was reviewed and approved by the Ethics Committees of Paris Lodron University of Salzburg (EK-GZ 26/2023) and Taras Shevchenko National University of Kyiv (11-22/7), and all participants provided documented written informed consent as part of the intro survey.

## Study design and procedure

This study was planned as a single-arm, open-label, uncontrolled pre-post evaluation study (see Fig 1). The study design consisted of three key assessment points: [1] baseline assessment (T0) conducted after one week of accommodation and adaptation, [2] mid-study engagement tracking (T1), which marked the beginning of the 6-week CBT-I intervention, and [3] post-intervention assessment (T2) conducted at the end of the program. Throughout the intervention, participants completed daily sleep diaries and, if available, recorded heart rate data.

Daily sleep quality data were assessed continuously between t0 and t2: every morning, participants filled in a sleep diary, where they reported how they felt during the night (subjective sleep quality on a scale from 0 to 10), how many times they thought they woke up during the night, how long these awakenings lasted, as well as the time they went to bed

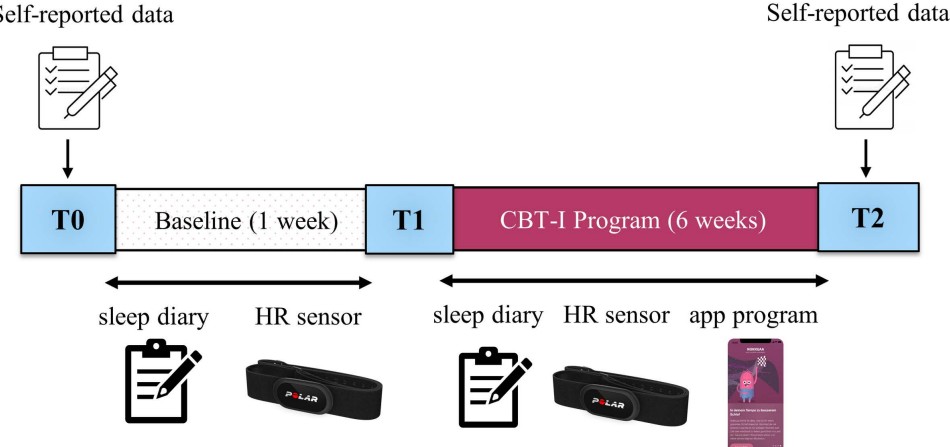

**Fig 1. 'Sleep2Ukraine' study design.** The effects of the CBT-I-based smartphone app (former NUKKUAA®, now Sleep²) on sleep were assessed subjectively via questionnaires (at T0 and T2) as well as a sleep diary every morning and objectively via a heart-rate sensor during the study. HR sensor – heart rate sensor.

and woke up. For participants who opted to use the heart rate HR sensor, this information was automatically retrieved and processed (see below for more details). Due to the fully anonymous mode of data collection, we could not prompt or encourage participants throughout intervention in case of missing diaries. Also, after intervention discontinuation at t2, we lost contact and could not conduct follow-up data, neither could we follow-up on dropouts.

## CBT-I-based app program

The Sleep2 app content was translated and adapted to fit the Ukrainian language, including adjustments to grammar, tone, and terminology to ensure cultural relevance and linguistic accuracy. This process involved ensuring the grammatical structures fit the Ukrainian language, adapting psychoeducational content to align with cultural beliefs about sleep, and replacing metaphors or examples unfamiliar to Ukrainian users with ones they would recognize. The program is organized into 6 consecutive levels, each featuring a set of components, based on core elements of CBT-I: various modes of communication (e.g., videos teaching sleep hygiene and psychoeducation), a chat-bot designed to help reframe thoughts, audio exercises for relaxation, tips on keeping good sleep habits, and blog posts discussing important scientific sleep topics (Table 1) [28].

This table summarizes the six levels of the Sleep2 app, each combining CBT-I components such as psychoeducational videos, relaxation exercises, chat-bot training, blog posts, and sleep hygiene advice to address sleep issues over a six-week period.

During the six-week app phase, participants were instructed to complete the program's six levels, which encompassed the most important contents of CBT-I. To reach level 6, participants were not required to complete 100% of the content within each level but were encouraged to engage with the core components. At t2, participants were redirected to post-treatment online questionnaires directly from the app. Participants who did not reach level 6 or complete the t2 measures were considered dropouts and excluded from the final analysis (we did not obtain information on partial completion, see Table 2 for details).

**Table 1. Overview over the contents of the app program divided by level [28].**

| Level | Video on sleep education | Auditory relaxation exercise | Sleep training via chat-bot | Blog | Sleep hygiene advice |
|---|---|---|---|---|---|
| 1 | Effects of tension and relaxation on sleep | Progressive muscle relaxation (PMR) | Topic: Rumination + exercise to putting those thoughts aside | 1. Correlations between sleep and depression/anxiety<br>2. Sleep and (cognitive) performance | "Go to bed only when you are tired" |
| 2 | Education on healthy sleep + sleep cycles | Imagination exercise | Topic: What to do with the "new free time" that was previously spent awake in bed | 1. Why do we need healthy sleep<br>2. Sleep, mental health, and online therapy | "After lying awake for 20 minutes, get up and return to your bed once you feel tired again" |
| 3 | Circadian rhythms | Breathing exercise | Topic: Sleep restriction | 1. Powering through the week and sleeping in at the weekend - a good idea?<br>2. Chronotype | "Only spend as much time in your bed as you'll actually sleep" |
| 4 | Factors influencing and disturbing healthy sleep | Imagination exercise | Topic: Habits influencing good sleep | 1. Insufficient sleep and impacts on health<br>2. Negative effects of electromagnetic radiation on sleep | "The bed should only be for sleeping (and sleeping with other people)" |
| 5 | Sleep and medication | Imagination exercise | Topic: Mindfulness | 1. Sleep medication<br>2. Natural sleep aids | "Try to have a regular sleep-wake-rhythm" |
| 6 | Sleep throughout the course of life | Meditation | Topic: Bedtime routines | 1. Age and sleep<br>2. Changes in sleep-wake-rhythms throughout the course of life | "Spend 30 minutes outside every day" |

**Table 2. Sociodemographic information, participant flow, engagement, and compliance.**

| Participants | N | Sociodemographics | | | Engagement (%) | Compliance (%) | | |
| | | Female sex (%) | Age (M±SD) | Financial Status (€) | *Moring Diaries* | *Exercises Completed* | *Educational Videos Watched* | *Chat-bot Completed* |
|---|---|---|---|---|---|---|---|---|
| Registered | 487 | 71 | | | | | | |
| Non-starters | 204 | 73 | | | | | | |
| Starters | 283 | 77 | 27.9±9.15 | 115.7 ± 89.2 | 35.68 | 51.27 | 8.89 | 8.21 |
| Non-completers | 123 | 77.7 | 29.2±10.27 | 116.4±112.1 | 30.41 | 37.69 | 7.26 | 6.13 |
| Completers | 160 | 76.9 | 27.3±8.14 | 123.02 ± 88.6 | 38.49 | 57.25 | 9.78 | 8.96 |
| No Sensor | 95 | 82.5 | 27.7±9.51 | 161.9±130.8 | 35.16 | 57.34 | 10.65 | 9.94 |
| With Sensor | 65 | 64.6 | 28.1±6.72 | 126.7 ± 99.9 | 39.63 | 54.53 | 7.88 | 7.51 |

## Material and measures

### Feasibility measures: Uptake, acceptance, engagement, and compliance

**Uptake.** Uptake was measured as the number of participants actually starting the intervention after initial sign up.

**Acceptance** was assessed using a 12-item self-report questionnaire specifically designed by the authors to address the program's primary goals and features, using a 10-point Likert-type scale ranging from 1 (strongly disagree) to 10 (strongly agree, see S1 Table). Participants were also given the opportunity to provide open-ended feedback regarding these points, with no restrictions. All questions were developed to align with previous findings emphasizing the importance of user engagement, clarity, and relevance in digital health interventions, particularly in culturally adapted CBT-I programs [28].

**Engagement** was monitored through the app's log data (frequency and duration of app use). Participants were encouraged to engage with the app via daily push-notifications.

**Compliance** was measured by the adherence to the program's requirements: participants were required to complete at least five audio exercises, one video, and one chatbot interaction per each of the 6 levels. We also measured the consistency of sleep diary entries, tracking whether participants regularly engaged with daily reporting as an indicator of sustained program involvement.

### Symptom measures

Self-reported data was obtained with online questionnaires at t0 and t2 focusing on a time range of one month prior to the date of filling out the questionnaire. Participants were required to respond to all questions, thus yielding no missing values. If no Ukrainian-adapted version of a questionnaire was available, items were translated using a standardized forward-backward translation procedure. This involved independent translations into Ukrainian and back-translations into the original language by separate bilingual experts. The research team then reviewed all items, resolved discrepancies through discussion, and ensured that the final version accurately reflected both linguistic and cultural nuances. For details on cutoff scores, reliability statistics, scoring ranges, and number of items, see S2 Table. Sociodemographic data were obtained on sex, age, employment status, working conditions, financial situation, and internally displaced persons (IDP) status (displacement status).

### Sleep-related questionnaires

- **Pittsburgh Sleep Quality Index (PSQI):** Subjectively perceived sleep quality was assessed using the 19-item Ukrainian version of the PSQI [29]. A global score greater than 5 was used as the cut-off to indicate poor sleep quality. This measure was operationalized as sleep problems because the PSQI's global score is widely interpreted as reflecting clinically significant sleep issues, encompassing factors such as sleep continuity, latency, and disruptions. Referring to these issues as "sleep problems" aligns with the interpretation of a global score >5 as indicative of sleep-related difficulties that affect overall sleep quality.

- **Insomnia Severity Index (ISI):** Insomnia symptoms were measured with the 7-item ISI [30]. A score above 7 was used as the cut-off for clinically significant insomnia symptoms.

- **Fear of Sleep Inventory (FoSI):** Fear of sleep in the past month was assessed using the 13-item FoSI [31]. A score above 15 was considered indicative of clinically relevant fear of sleep.

### Mental health symptom questionnaires

- **General Anxiety Disorder-7 Scale (GAD-7):** Anxiety symptoms were measured with the Ukrainian version of the GAD-7 [32]. A cut-off score of 5 or greater was used to indicate clinically significant anxiety.

- **Patient Health Questionnaire-9 (PHQ-9):** Depression symptoms were measured using the Ukrainian version of the PHQ-9 [33]. A score above 5 indicated clinically significant depressive symptoms.

- **PTSD Checklist for DSM-5 (PCL-5):** PTSD symptoms were assessed with the 20-item Ukrainian version of the PCL-5 [34]. A score above 31 was used to identify clinically significant PTSD symptoms.

- **Perceived Stress Scale (PSS-4):** Perceived stress was measured with the 4-item PSS-4 [35]. A cut-off score of 7 or greater indicated elevated stress levels.

- **Somatic Symptom Scale (SSS-8):** Somatic symptoms were assessed using the 8-item SSS-8 [36]. Scores above 4 were used as the cut-off for clinically significant somatic symptoms.

### Objective sleep measurement

In participants using the Polar® H10 heart rate sensor (Polar Electro GmbH Deutschland), continuous objective sleep measures were collected. The device recorded inter-beat intervals (IBIs), which were securely transmitted each morning along with subjective sleep diaries via HTTPS to a central server. These IBIs were processed using a multi-resolution convolutional neural network (MCNN), trained on a large dataset of manually scored polysomnography (PSG) recordings (~9,000 nights). The MCNN classified each 30-second epoch into one of four sleep stages (wake, light sleep [N1 + N2], deep sleep, or REM) based solely on heart rate variability patterns, achieving accuracy comparable to expert inter-rater reliability ($\kappa = 0.69$). Preprocessing steps included artifact detection and corrections to ensure data quality, and from the resulting hypnograms, sleep parameters such as total sleep time (TST), time in bed (TIB), sleep onset latency (SOL), wake after sleep onset (WASO), number of awakenings (NOA), and sleep efficiency (SE) were extracted [25,26]. Sleep stages were not reported because heart rate-based classification provides individualized sleep stage data that varies significantly between participants. Since there is no standardized way to aggregate or compare sleep stages across individuals without introducing inaccuracies, we focused on broader, validated sleep parameters such as sleep onset latency, number of awakenings, and total sleep time. Instead of reporting aggregated sleep stage data, participants received personalized feedback on their sleep patterns based on their individual nightly variations.

## Statistical analysis

Daily/nightly data were averaged over the baseline (first 7 days of the study between t0 and t1) and over the last 7 days before the end of the treatment (t2, see Fig 1) for statistical analysis, as performed in R, version 2023.03.0 [37]. Extreme outliers, that is, data points deviating more than ±2 $SD$ from the mean, were excluded from the respective analysis. Besides completer analyses we also report intention-to-treat (ITT) analyses (with pre-assessment values (t1) were carried forward to the post-assessment (t2)).

## Inclusivity in global research

Additional information regarding the ethical, cultural, and scientific considerations specific to inclusivity in global research is included in the Supporting Information (S1 Checklist).

# Results

## Program uptake

A total of 487 people initially registered for the sleep treatment, out of which 283 started (t0) the program, and 160 finished (t2): 95 without continuous objective measurements and 65 with the sensor (see Table 2) resulting in a completion rate (t0-t2) of 56.54%. There were no significant differences between starters and completers on sociodemographics, but completers evidently completed more content. The analysis of the dropouts (N = 123) in comparison to program completers (N = 160) revealed no difference in age ($t(250)$ = 0.45, $p$ =.651), sex ($x^2(3)$ = 2.96, $p$ =.398), employment status ($x^2(1)$ = 0.25, $p$ =.619), working conditions ($x^2(2)$ = 0.37, $p$ =.829), displacement status ($x^2(3)$ = 2.52, $p$ =.472), and financial situation ($x^2(7)$ = 8.78, $p$ =.269). Interestingly, participants who requested and received the HR sensor were more likely to be male and had a less favourable financial situation. Yet, critically, they were not consistently more engaged with the program.

The table represents average values per every group of participants. Compliance is calculated based on the total number of entries from a single participant of daily audio exercises completed (had to be performed minimum one time per day), psychoeducational videos and chat (had to be performed at least once per level).

At t0, completers reported high levels of sleep problems ($M$ = 8.93, $SD$ = 3.45), low to medium levels of insomnia ($M$ = 11.98, $SD$ = 5.65), anxiety ($M$ = 7.36, $SD$ = 3.81), depression ($M$ = 11.82, $SD$ = 5.48), PTSD symptoms ($M$ = 27.27, $SD$ = 15.21), perceived stress ($M$ = 7.38, $SD$ = 3.18), and somatic symptoms ($M$ = 9.42, $SD$ = 4.73), as well as low levels of fear of sleep ($M$ = 5.55, $SD$ = 6.67; see Table 3). These baseline values confirm that we successfully reached our target population. On average, 6.36 ($SD$ = 13.51) out of 40 nights were disturbed by air-raid alarms.

The table presents mean values and standard deviations for all measured symptoms, together with the clinical cutoffs per every measure, number of participants and corresponding percentage.

## Engagement: session completion and overall app use

Overall, the N = 160 completers saved 80.72% of the 50 required sleep diaries and used the app on an average of 39.8 days ($SD$ = 19.12) out of a maximum of 50 days, indicating good engagement with the app.

**Acceptance and program usability.** Participants were generally satisfied with the treatment and the app, particularly with content clarity, relevance, relaxation exercises, and program structure. However, several aspects received lower ratings, such as chatbot responses, push notifications, and perceived sleep improvement. These results indicate that while the program was well-received overall, certain features require refinement to enhance user experience and engagement (see Table 4).

The table presents the average survey results on the acceptance of the proposed program, sorted in descending order. Participants rated the app and the sensor on specified criteria using a 10-point Likert scale (1 = strongly disagree to

**Table 3. Summary of the screened symptoms at baseline (t0) for completers (n=160).**

| Symptom | Mean | SD | Cutoff | Count | Percentage |
|---|---|---|---|---|---|
| Sleep Problems | 8.93 | 3.45 | No sleep problems | 16 | 9.69 |
| | | | Sleep problems | 149 | 93.31 |
| Insomnia | 11.98 | 5.65 | No symptoms | 33 | 23.03 |
| | | | Subthreshold insomnia | 74 | 44.85 |
| | | | Clinical insomnia of moderate severity | 46 | 27.88 |
| | | | Severe clinical insomnia | 7 | 4.24 |
| Fear of Sleep | 5.55 | 6.67 | No symptoms | 130 | 81.81 |
| | | | Mild fear | 21 | 12.73 |
| | | | Moderate fear | 8 | 4.85 |
| | | | Severe fear | 1 | 0.61 |
| Somatic Symptoms | 9.42 | 4.73 | No symptoms | 14 | 11.52 |
| | | | Minimal symptoms | 15 | 9.09 |
| | | | Low symptoms | 45 | 27.27 |
| | | | Medium symptoms | 55 | 33.33 |
| | | | Severe symptoms | 31 | 18.79 |
| Anxiety | 7.36 | 3.81 | No symptoms | 41 | 27.88 |
| | | | Mild anxiety | 76 | 46.06 |
| | | | Moderate anxiety | 43 | 26.06 |
| | | | Severe anxiety | 0 | 0 |
| Depression | 11.82 | 5.48 | No symptoms | 8 | 7.89 |
| | | | Mild depression | 48 | 29.09 |
| | | | Moderate depression | 52 | 31.51 |
| | | | Moderately severe depression | 33 | 20 |
| | | | Severe depression | 19 | 11.51 |
| PTSD | 27.27 | 15.21 | No symptoms | 103 | 65.46 |
| | | | Presence of PTSD | 10 | 6.06 |
| | | | Severe PTSD | 47 | 28.48 |
| Perceived Stress | 7.38 | 3.18 | No stress | 65 | 42.43 |
| | | | Increased stress | 95 | 57.57 |

10 = strongly agree). The evaluation intervals are as follows: very bad (0-2), bad (3-4), mediocre (5-6), good (7-8), very good (9-10). Formulations of items is presented in S1 Table. Results are presented for the sensor users (N = 65).

As for **qualitative feedback**, 38% of sensor users (25 out of 65) reported that the sensor restricted sleeping positions, was uncomfortable to adjust and wear, or caused some degree of discomfort. Additionally, some participants mentioned occasional synchronization and battery issues. While many found the objective sleep analysis helpful (78%), others doubted its accuracy (12%). Many (56%) participants preferred a wrist/arm sensor instead of a chest belt. Positive feedback highlighted ease of use, diverse content, helpful relaxation exercises (especially before sleep and after air raid alarms), pleasant design, simple interface, engaging educational content and chatbot, and the ability to take and reflect on notes within the app. Negative feedback included limited exercise variety, inability to revisit or list all exercises, lack of video adjustments (speed, narrator, subtitles), no late sleep entries or nap recordings, limited chatbot responses, and dissatisfaction with the level structure with an emphasis on the preference for a topic-based division instead.

**Table 4. Summary of the program evaluation.**

| Category | Mean | SD |
|---|---|---|
| Content clarity | 8.69 | 1.88 |
| Recommend Sleep2Ukraine to others | 8.17 | 2.07 |
| Content relevancy | 7.64 | 2.2 |
| Usefulness of relaxation exercises | 7.6 | 2.68 |
| Levels relevancy | 7.22 | 2.52 |
| Convenience to sleep | 7.03 | 2.47 |
| Improved sleep | 6.49 | 2.27 |
| Usefulness of imaginary exercises | 6.69 | 2.99 |
| Sleepscore accuracy | 6.67 | 2.43 |
| Easy to meet requirements for next level | 6.38 | 2.84 |
| Personalized program effectiveness | 6.25 | 2.29 |
| Usefulness of push-notifications | 5.8 | 3.25 |
| Usefulness of the chatbot | 5.38 | 3.14 |
| Problems using sensor | 4.99 | 3.13 |
| Wake up because of sensor | 3.01 | 2.86 |

**Table 5. Changes in mental health symptoms after the sleep intervention (n = 160 completers).**

| | T0 | T2 | T2-T0 | t(159) | p | Cohen's d |
|---|---|---|---|---|---|---|
| | M ± SD | M ± SD | | | | |
| Sleep Problems | 8.93 ± 3.45 | 6.91 ± 3.02 | -2.02 | 6.64 | <.001 | .53 |
| Insomnia | 11.98 ± 5.65 | 7.78 ± 4.84 | -4.20 | 8.75 | <.001 | .69 |
| Fear of Sleep | 5.55 ± 6.67 | 3.75 ± 5.83 | -1.80 | 3.12 | .002 | .25 |
| Anxiety | 7.36 ± 3.81 | 5.34 ± 3.21 | -2.02 | 6.05 | <.001 | .48 |
| Depression | 11.82 ± 5.48 | 8.43 ± 4.74 | -3.39 | 6.52 | <.001 | .52 |
| PTSD | 27.27 ± 15.21 | 18.42 ± 13.97 | -8.85 | 6.34 | <.001 | .51 |
| Stress | 7.38 ± 3.18 | 6.06 ± 2.74 | -1.32 | 4.92 | <.001 | .39 |
| Somatic Symptoms | 9.42 ± 4.73 | 7.10 ± 4.19 | -2.32 | 6.34 | <.001 | .51 |

## Pre-post changes on sleep and mental health

Completers reported statistically significant improvements on all symptoms after the sleep intervention. Specifically, sleep problems decreased by 22.60% (from $M = 8.93$ to $M = 6.91$), insomnia by 35.08% (from $M = 11.98$ to $M = 7.78$), fear of sleep by 32.43% (from $M = 5.55$ to $M = 3.75$), anxiety by 27.72% (from $M = 7.36$ to $M = 5.34$), depression by 28.67% (from $M = 11.82$ to $M = 8.43$), PTSD symptoms by 32.41% (from $M = 27.27$ to $M = 18.42$), somatic symptoms by 24.52% (from $M = 9.42$ to $M = 7.10$), and perceived stress by 17.90% (from $M = 7.38$ to $M = 6.06$). Effect sizes ranged from small to medium (see Table 5 and S1 Fig). However, post-treatment values for insomnia, PTSD, and depression remained above clinical cutoffs for a substantial proportion of participants: specifically, 41.82% still reported clinically significant insomnia (ISI > 7), 32.42% remained above the PTSD threshold (PCL-5 > 31), and 39.39% continued to show depressive symptoms above the clinical cutoff (PHQ-9 > 5). This indicates that while improvements were evident, a notable portion of participants continued to experience clinically relevant symptoms. Similar results were found in the ITT analysis (N = 283; see S3 Table), which speaks against selective dropout effects.

The table presents the results of paired-sample t-test for subjective measures obtained with the questionnaires at T0 and T2 (see Fig 1 for details). Results are presented for program completers (N = 160).

**Table 6. Changes in continuous sleep measures from pre (t1) to post (t2) intervention.**

|  | T1 | T2 | T2-T1 | t(73) | p | Cohen's d |
|---|---|---|---|---|---|---|
|  | M ± SD | M ± SD |  |  |  |  |
| TIB | 492.45 ± 90.37 | 489.69 ± 82.77 | 4.84 | 0.891 | .376 | .11 |
| TST | 427.17 ± 77.95 | 428.74 ± 80.26 | -0.91 | -0.180 | .858 | -.02 |
| SOL | 29.71 ± 33.34 | 25.01 ± 29.28 | 4.71 | 2.047 | .044 | .24 |
| NOA | 2.44 ± 2.45 | 2.42 ± 2.35 | 0.06 | 0.403 | .688 | .05 |
| WASO | 40.49 ± 33.69 | 40.56 ± 36.84 | 0.66 | 0.205 | .838 | .02 |
| SE | 86.93 ± 8.03 | 87.66 ± 8.63 | -0.83 | -1.368 | .176 | -.16 |

### Objective sleep measures (sensor users only)

Table 6 shows that the N = 65 program completers with sensors demonstrated a statistically significant reduction in objective SOL, with the time to fall asleep decreasing from approximately 29 to 25 minutes. No other significant effects were observed on the remaining objective sleep measures.

NOA (number of awakenings) here is a count of awakenings of more than 2 continuous minutes. TIB – time in bed, TST – total sleep time, SOL – sleep onset latency, WASO – wake after sleep onset, SE – sleep efficiency. Results are presented for sensor users (N = 65).

## Discussion

To our knowledge, this is the first study to report on a non-pharmacological, digital CBT-I intervention during an ongoing war. The results, obtained from Ukraine, demonstrated high uptake/feasibility and acceptance of the program, with significant improvements across all sleep and mental health measures, even though the findings remain preliminary due to the study's uncontrolled design. We will discuss each of these in this order in the following.

### Uptake, feasibility, acceptance, usability

*Program uptake* was acceptable: out of 487 registered users, 283 started. Out of 283 starters 160 completed, thus, a completion rate was 56%, which can be considered good for an unguided program, given the circumstances and the high demands of the program (6 levels, 50 required diaries). This means that recruitment for the subsequent randomized controlled trials (RCT) should be very feasible. Interestingly, despite additional logistics via mail for the sensor and the discomfort reported by some, 65 out of the 160 program completers proceeded with the daily use of the HR sensor.

Regarding *engagement*, participants used the app for a duration of 6 weeks, with 57% completing the full treatment and showing above-average compliance and engagement. The overall dropout rate of 43.46% is similar to other studies investigating the effects of digital programs for insomnia in outpatients: 40% [38] and 34.4% [39]. In fact, most participants completed at least half of the program (~20 days), with no obvious demographic differences between dropped-out participants and program completers.

Although objective feedback on sleep parameters is not required for diagnosing or treating insomnia, some participants reported it as helpful in raising awareness and motivating behavioral change. Nevertheless, insomnia remains a subjective condition, and objective assessments such as heart rate-based sleep tracking are generally recommended only when other sleep disorders are suspected. Importantly, the delivery and daily use of HR sensors pose a logistical hurdle that could limit scalability. Given our non-randomised assignment to sensor vs. non-sensor groups we can only hint at these effects. Sensor users were more likely to be male and had less favourable financial situation. The further might reflect some gender-stereotypical technology affinity in Ukraine users, whereas the latter points to the potentially higher subjective value of receiving a costly wearable. Importantly, we cannot conclude that the sensor users were more engaged.

Should this be confirmed in further studies with random assignment to sensor vs. no-sensor groups then we could remove this hurdle to large scale application (at the cost of objective sleep data).

*User satisfaction and potential app improvements*. Those who completed the treatment and moved to t2 questionnaires were highly satisfied, especially with the (culturally adapted) intervention content, relaxation exercises, assigned levels, and were willing to recommend the program. The lowest scores were for the chatbot and push notifications, indicating these features were less effective. As for qualitative feedback, only some sensor users reported significant discomfort. Objective feedback was considered useful, with positive remarks on relaxation exercises and negative feedback on their limited variation and inability to revisit previous exercises. Participants also disliked the inability to record daytime naps and/or sleep shorter than four hours. However, this is a technical limitation as the algorithm has not been trained on nap data and, therefore, these activities could not have been included into the program, as well as into the analysis.

The positive reception of the intervention and the relatively low dropout rate suggest that there is a significant demand and openness to internet-based interventions among people in Ukraine. This contrasts with the situation in Western countries, where internet interventions often face challenges with low uptake [40,41]. One possible reason for this difference is the low availability of mobile interventions in the Ukrainian language. This indicates a general reluctance among Ukrainians to use interventions available in English or other languages. This insight has important implications for the development of interventions in other mental health areas for Ukraine: language accessibility appears to be a crucial factor.

### Sample characteristics, pre-post effects on sleep parameters and mental health

**Participant flow and symptom severity.** Regarding the pathology of the sample and whether we addressed the program's target group: our sample demonstrated significant sleep impairments, confirming that we reached the intended population – only 9% of the completers (14 out of 160) did not report sleep problems at baseline, meaning that even these individuals chose to complete the full program, potentially seeking general sleep improvement or stress reduction. Before starting the intervention, participants who completed the program exhibited significantly elevated levels of sleep problems. Specifically, 60% of completers scored above the clinical cutoff on the PSQI (score >5; $M = 8.93$, $SD = 3.45$), indicating clinically relevant sleep difficulties. Moreover, 30% had PSQI scores greater than 10, which is often interpreted as indicating more severe or chronic sleep problems. Regarding insomnia symptoms, 44% of participants met criteria for subthreshold insomnia and 27% for clinical insomnia based on the ISI, with scores above 7 ($M = 11.98$, $SD = 5.65$). These rates are nearly three times higher than typically observed in Western population studies, where while up to 30–48% of adults may report some insomnia symptoms, only 8.5–13% meet criteria for clinical insomnia with daytime consequences, and just 6–15% receive an actual diagnosis [28,42]. Our study did not exclude participants based on the presence or severity of sleep problems. However, given the overall low number of individuals without sleep issues, we can conclude that their limited representation likely did not significantly influence the effect sizes observed in the results. We also observed very high levels of severe PTSD among participants, potentially reflecting the ongoing air war on civilians, which highlights the necessity for PTSD-specific content in the program. In addition to this, more than 52 percent of our participants had elevated somatic symptoms, 60 percent mild to moderate anxiety or depression, and more than half reporting elevated levels of stress.

**Pre-post effects on mental health symptoms and sleep parameters.** Program completers reported substantial reductions in subjective sleep problems (minus 22.60%), insomnia severity (minus 35.08%), and fear of sleep (minus 32.43%), all with low to medium-large effect sizes. There were also notable decreases in anxiety (minus 27.72%), depression (minus 28.67%), PTSD (minus 32.41%), somatic symptoms (minus 24.52%), and perceived stress (minus 17.90%). However, compared to traditional face-to-face CBT-I interventions, the effect sizes in our study appear somewhat smaller [43]. This may be due to several factors, including the lower baseline severity of insomnia in our sample, the unguided nature of the intervention, and the absence of personalized therapist support. Additionally, the high psychological burden of ongoing war conditions may have limited participants' ability to fully engage with and benefit from the intervention, thus reducing its overall effectiveness. Future studies should explore ways to enhance treatment engagement, such as

incorporating tailored motivational support or hybrid approaches combining digital and human guidance. In sensor-wearing completers, objective sleep measurements did not show large scale significant changes post-treatment, except for a slight reduction in objective sleep onset latency (SOL). Given that the baseline objective sleep data indicated no significant sleep problems (with total sleep time above 7 hours), the lack of objective changes over the 6-week study period is not surprising. The striking discrepancy between subjective and objective sleep measures is a well-documented phenomenon, particularly in insomnia, which is fundamentally a subjective disorder. Insomnia is defined by a person's self-reported difficulties falling or staying asleep and the associated daytime impairment, regardless of whether objective measurements (e.g., actigraphy or polysomnography) confirm these problems. This often leads to a mismatch between perceived and physiologically measured sleep – a pattern referred to as sleep-wake state misperception or subjective-objective sleep discrepancy [44]. Thus, our findings reiterate the predominantly subjective nature of many sleep disorders [45].

Our results align with previous findings [46], highlighting the strong co-occurrence between sleep problems and somatic symptoms [47], depression and anxiety [7,48], and PTSD [49]. A main finding of our study shows that all trauma-related disorders improve over the course of a predominantly sleep-focused program. This might be related to the above-mentioned indirect pathways: as sleep recovers, also depression and PTSD might do so. Yet, many of the taught CBT-based cognitive techniques are applicable to these non-sleep related symptoms just as well. Further, self-efficacy related to sleep improvements might translate onto other domains as well.

When analyzing data from additional participants not included in this publication (in response to reviewer comments), we observe that participants with more severe initial symptoms showed better treatment effects, which aligns with findings that symptom severity can predict treatment efficacy [50].

## Limitations and future directions

Our study has several limitations. First, it used a convenience sample and lacked a control group, as it was designed as a single-arm, open-label, uncontrolled pre-post evaluation study. Future studies with randomized group assignment and the inclusion of control groups are necessary to rule out that the observed improvements are not solely attributable to factors such as the passage of time, measurement repetition, increased symptom awareness, treatment expectancies [43], or other external influences. However, in a war-torn population, the ethical challenges of using an inactive control or waitlist group are substantial. Additionally, the fully anonymous data collection in this study precluded follow-up assessments.

Secondly, while we previously suggested that participants who self-subscribed might have had more severe sleep and mental health issues, the data indicates otherwise: on average, participants reported low to medium levels of insomnia and increased level of sleep problems as assessed by the ISI and PSQI respectively. This suggests that the program attracted individuals with varying levels of need, including those not experiencing severe symptoms. Additionally, the recruitment strategy may have introduced selection bias by appealing more to younger individuals who are more comfortable with technology. This limits the generalizability of our findings to older populations who may not engage as readily with mobile-based interventions.

Despite these limitations, scalable mobile-based interventions remain crucial for the Ukrainian population, particularly in war-threatened areas with internet and electricity access. Our study suggests that such interventions can be delivered and accepted, but findings are preliminary due to the study design. Further research, including randomized controlled trials, is needed to establish the efficacy of mobile-based CBT-I in Ukraine.

## Supporting information

**Supplementary Tables 1 to 3. Formulations of items regarding the acceptance of the program.** This table lists the questions used to evaluate participants' acceptance of the program, including perceived effectiveness, clarity of content, usefulness of exercises, and likelihood of recommending the program.
(DOCX)

**Supplementary Fig 1. Density differences between pre- and post-measurements for program completers (N = 160).** This figure illustrates the density differences in subjective measure scores between pre- and post-program assessments for participants who completed the program. The red vertical line indicates the clinical cutoff. Percentages of participants in clinical and non-clinical categories pre- and post-intervention are shown.
(TIF)

## Acknowledgments

IBI-based algorithms for sleep classification were kindly provided for scientific purposes by sleep² (Nukkuaa GmbH).

## Author contributions

**Conceptualization:** Anton Kurapov, Jens Blechert, Alexandra Hinterberger, Manuel Schabus.

**Data curation:** Anton Kurapov.

**Formal analysis:** Anton Kurapov.

**Funding acquisition:** Anton Kurapov.

**Investigation:** Anton Kurapov, Manuel Schabus.

**Methodology:** Anton Kurapov, Jens Blechert, Alexandra Hinterberger, Manuel Schabus.

**Project administration:** Jens Blechert, Manuel Schabus.

**Resources:** Jens Blechert, Manuel Schabus.

**Software:** Alexandra Hinterberger, Manuel Schabus.

**Supervision:** Jens Blechert, Manuel Schabus.

**Writing – original draft:** Anton Kurapov.

**Writing – review & editing:** Jens Blechert, Alexandra Hinterberger, Pavlos Topalidis, Manuel Schabus.

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
