## [Decision Letter · Decision Letter 0]

4 Dec 2024

Dear Dr. Kurapov,

Thank you for submitting your manuscript to PLOS ONE. After careful consideration, we feel that it has merit but does not fully meet PLOS ONE’s publication criteria as it currently stands. Therefore, we invite you to submit a revised version of the manuscript that addresses the points raised during the review process.

The manuscript has been reviewed by two referees, and we are requesting a major revision before it can be considered for acceptance. We kindly ask you to provide a revised version of the manuscript, specifically:

Revise the abstract and introduction as per the reviewers’ suggestions to improve clarity and provide additional background as needed.Provide methodological clarifications, including details on the study design, recruitment, inclusion/exclusion criteria, and components of the CBT-I intervention.Update the results and discussion sections based on the reviewers’ recommendations, addressing issues such as baseline severity, effect size interpretations, and generalizability of findings.

We look forward to receiving your revised manuscript.

Kind regards,

Serena Scarpelli

Academic Editor

PLOS ONE

“Research was funded by MSCA4Ukraine, a funding scheme that is implemented by a consortium comprised of Scholars at Risk Europe (SAR Europe) hosted at Maynooth University, Ireland (project coordinator), the German Alexander von Humboldt Foundation (AvH) and the European University Association (EUA); grant number 1233157.

IBI-based algorithms for sleep classification were kindly provided for scientific purposes by sleep² (Nukkuaa GmbH).”

“MSCA4Ukraine.”

3. Please include a complete copy of PLOS’ questionnaire on inclusivity in global research in your revised manuscript. Our policy for research in this area aims to improve transparency in the reporting of research performed outside of researchers’ own country or community. The policy applies to researchers who have travelled to a different country to conduct research, research with Indigenous populations or their lands, and research on cultural artefacts. The questionnaire can also be requested at the journal’s discretion for any other submissions, even if these conditions are not met.  Please find more information on the policy and a link to download a blank copy of the questionnaire here: https://journals.plos.org/plosone/s/best-practices-in-research-reporting. Please upload a completed version of your questionnaire as Supporting Information when you resubmit your manuscript.

“I have read the journal's policy and the authors of this manuscript have the following competing interests: Manuel Schabus is Co-Founder and CSO of NUKKUAA®”

7. We note that you have referenced (Kurapov A, Spanhel K, Schabus M. Nukkuaa4Ukraine Project: Cultural Adaptation of Cognitive Behavioral Therapy of Insomnia for Ukrainians during the War. Unpublished manuscript Department of Psychology, Paris Lodron University of Salzburg. 2024) which has currently not yet been accepted for publication. Please remove this from your References and amend this to state in the body of your manuscript: (ie “Bewick et al. [Unpublished]”) as detailed online in our guide for authors http://journals.plos.org/plosone/s/submission-guidelines#loc-reference-style

8. We notice that your supplementary tables are included in the manuscript file. Please remove them and upload them with the file type 'Supporting Information'. Please ensure that each Supporting Information file has a legend listed in the manuscript after the references list.

Reviewers' comments:

Reviewer's Responses to Questions

**Comments to the Author**

1. Is the manuscript technically sound, and do the data support the conclusions?

Reviewer #1: Partly

Reviewer #2: Partly

2. Has the statistical analysis been performed appropriately and rigorously?

Reviewer #1: Yes

Reviewer #2: Yes

3. Have the authors made all data underlying the findings in their manuscript fully available?

Reviewer #1: Yes

Reviewer #2: Yes

4. Is the manuscript presented in an intelligible fashion and written in standard English?

Reviewer #1: Yes

Reviewer #2: Yes

Reviewer #1: The article investigates the efficacy of a digital CBT-I program in Ukraine sample. The study is interesting and has relevant potential in its context. Nevertheless, I provide several suggestions to implement the readability of the manuscript and contents. The authors should organize the manuscript more clearly; in several point it is not very clear and should deepen some sleep-related aspects.

Full and short title is the same. It is supposed to be different.

In the abstract the background section lacks, and it is not really understandable. The authors should rewrite it in a more linear way.

Introduction: I suggest removing the subheadings.

Line 98-99: please specify the sleep parameters and mental health indices.

Line 138: the first paragraph of method section should be the participants section, in which it should be described participants inclusion and exclusion criteria and sample characteristics.

Line 141: specify which are the t0, t1 and t2 measures and how long does it take each t. Furthermore, even if the study plan is described in Figure 1, I suggest describing the study design more deeply in the text. For example, a brief description- even in bullet points- of the time intervals (t0, t1 and t2) and the relative measures acquired at each point.

Line 191 and 196: I suggest explaining deeply the questionnaire and to do it in bullet point, instead of bold. It could be more easily to the reader to understand. Furthermore, the PSQI does not measure sleep disturbances but subjectively perceived sleep quality. Did the authors control for cutoff scores? On what bases sample is divided in sleep disturbances and chronic sleep problems? Authors should better describe the sample data regarding sleep and health intext, describing score obtained at t0 and at t2, if reached scores under the cutoff levels.

For health data (table 2) no participants have scores under the cutoff?

Line 205: explain how the HR works and how it works in relation to sleep.

Line 215: explain/describe daily/night data.

Line 279: specify in the text if the improvement of symptoms reached to clinical values.

Line 292: specify if SOL reduction is statistically significant, and the other measures?

The authors should specify and describe in the text the CBT-I components comprised in the app.

Reviewer #2: General Comments

This manuscript reports results of a single arm, uncontrolled clinical trial assessing the acceptability and efficacy of an unguided mobile app offering cognitive-behavioral therapy for insomnia (CBT-I) in Ukrainian people experiencing the ongoing war. The authors should be commended for having undertaken such an innovative study on a significant need of this population who experience frequent nighttime events (e.g., air-raid alarms) likely to impair their sleep, on top of war-related anxiety also likely to negatively impact sleep. The study has the typical limitations of an uncontrolled clinical trial, which are however, well acknowledged by the authors. Although I am enthusiastic about this manuscript, I recommend making a number of revisions to clarify some aspects of the study methodology and to come up with an interpretation that better applies to the findings.

Specific Comments

Abstract

1. It would be nice to state in the abstract that there were 160 completers and to specify the proportion that this number represents on the total number of starters.

2. I suggest making it clearer that the objective measure of sleep was provided by the ambulatory heart rate sensor, as this is one innovation of this study.

3. As I will detail later, I disagree with the conclusion that effect sizes for subjective measures were all in the medium to high range. I suggest revising this and adding the actual range of effect sizes obtained.

4. I don’t agree with the conclusion that “These findings demonstrate the potential of scalable mobile-based CBT-I…with or without the instrument, based on the heart rate assessment”. This aspect of the intervention was among the least appreciated by the study participants. I would rephrase this to take that finding into account.

Introduction

• What is unclear, and could be clarified, is if this study is the first to assess the efficacy of CBT-I, regardless of the treatment format, in people facing war.

• What does “low threshold” mean when describing the advantages of a mobile-based CBT-I?

• Given that heart rate is described as able to infer sleep stages, I find it surprising that sleep stages data were not reported in this manuscript.

• High uptake, feasibility and acceptance were hypothesized but are not operationalized. Were criteria to conclude about “high levels” determined a priori, before the analysis?

Methods

• There is no mention of the recruitment method used. In addition, what were the inclusion and exclusion criteria? These are not mentioned. This is important in order to know to what extent results may be generalized to the whole population.

• Also, it seems that the presence of clinically meaningful sleep problems was not required for participation. This needs to be taken into account when interpreting the data. One may not expect the same magnitude of effect sizes if a significant proportion of participants were non-clinical cases at baseline.

• Was the content of CBT-I adapted to this population or was it a generic CBT-I?

• It is unclear what acceptance tool the authors adapted to assess acceptance in this study.

• Several criteria are stated to identify completers (e.g., completed five audio exercises, etc.). However, what proportion of the total content do these represent?

• Could the authors explain why they chose an anonymous collection of data?

• One weakness is the translation of the measurement tools by the authors. A more systematic translation approach would have been needed.

• Could the authors explain briefly Topalidis et al.’s method to analyse the HR data? They refer to another publication but, as this method is not well known, I suggest explaining it briefly.

• For each measure, the authors need to state what cut-off scores they used to determine the presence of clinical levels of symptoms.

• Please specify how the qualitative feedback was collected and among how many participants.

Results

• Rather than saying, “there were no obvious differences…”, please rephrase as follows: “there were no significant differences…” (p. 9, 3rd paragraph).

• What was the variable “displacement status” and how was that operationalized?

• The mean ISI score at baseline was not that elevated. The impact of this on results obtained should be discussed.

• In Table 3, several mean scores fall in the “mediocre” range. Yet, the authors conclude that “most of the participants were very satisfied with the treatment and the app. I think it is important to tone down a bit that kind of interpretation.

• Please clarify this phrase: “…a discouraging level structure suggesting continuous sleep improvement” (p. 12, 1st paragraph).

• Please use Cohen’s criteria for characterizing the effect sizes obtained: small (d=0.2), medium (d=0.5) and large (d=0.8). As such, effect sizes are not in the “medium to high” range but medium at best. Please change the findings interpretation accordingly.

• It would be important to note that the PSQI score was still in the clinical range at T2.

• Please specify that the difference obtained on objective SOL was significant and that no other significant effect was found on objective sleep measures.

Discussion

• Sometimes the authors describe their intervention as being a “training”. I would rather use the terms “intervention” or “treatment”.

• Given the small effect obtained on fear of sleep, should the intervention target this aspect more specifically? How so?

• The authors emphasize the importance of targeting PTSD-specific content in the program, yet they did not mention whether or not that was included in their program.

• One study limitation that is discussed is the possibility that participants who self-subscribed might have had more severe insomnia resulting in a selection bias. Personally, I don’t find this is a problem. As mentioned earlier, participants had on average low levels of insomnia and sleep disturbances as assessed with the ISI and the PSQI. Hence, this does not appear to be true. Besides, this is what we want when we develop an intervention, to attract patients who are the most likely to benefit from it, in other words, those who have more severe problems.

• What appears most important to emphasize to me in terms of selection bias is the fact that an app may have appealed more to younger individuals, i.e., those who are more at ease with technology, thus limiting the generalization of findings to older individuals.

**Do you want your identity to be public for this peer review?** For information about this choice, including consent withdrawal, please see our Privacy Policy

Reviewer #1: No

Reviewer #2: No

---

## [Author Response · Author response to Decision Letter 1]

28 Dec 2024

Response has been provided as a separate file

---

## [Decision Letter · Decision Letter 1]

29 Jan 2025

Dear Dr. Kurapov,

Thank you for submitting your manuscript to PLOS ONE. After careful consideration, we feel that it has merit but does not fully meet PLOS ONE’s publication criteria as it currently stands. Therefore, we invite you to submit a revised version of the manuscript that addresses the points raised during the review process.

We look forward to receiving your revised manuscript.

Kind regards,

Serena Scarpelli

Academic Editor

PLOS ONE

Journal Requirements:

Reviewers' comments:

Reviewer's Responses to Questions

**Comments to the Author**

Reviewer #2: (No Response)

Reviewer #3: (No Response)

2. Is the manuscript technically sound, and do the data support the conclusions?

Reviewer #2: Partly

Reviewer #3: Yes

3. Has the statistical analysis been performed appropriately and rigorously?

Reviewer #2: Yes

Reviewer #3: (No Response)

4. Have the authors made all data underlying the findings in their manuscript fully available?

Reviewer #2: Yes

Reviewer #3: (No Response)

5. Is the manuscript presented in an intelligible fashion and written in standard English?

Reviewer #2: No

Reviewer #3: (No Response)

Reviewer #2: General Comments

This is a resubmission of a manuscript reporting results of a single arm, uncontrolled clinical trial assessing the acceptability and efficacy of an unguided mobile app offering cognitive-behavioral therapy for insomnia (CBT-I) in Ukrainian people experiencing the ongoing war. The revised version has much improved but has raised other issues that, in my opinion, need to be clarified. A few other issues are not well addressed. Moreover, I think that a linguistic revision is needed as I detected several grammatical errors.

Specific Comments

Introduction

1. A better expression for “low threshold”, that does not need to be defined, is “low intensity” intervention.

2. High uptake, feasibility and acceptance were hypothesized but are still not operationalized. The authors mention that these don’t need to be defined in pilot studies, a statement with which I disagree. If no criteria is stated a priori, then it is hard to conclude whether the intervention is or is not feasible and well accepted.

3. In what language and for what kind of population was Sleep2 initially developed?

Methods

4. The description of the time points (i.e., study design) in the text is still unclear to me. Please rework that section for more clarity.

5. I don’t understand this : “…sleep stages were not reported due to their limited generalizability”. Please clarify.

Results

6. In Table 4, several mean scores fall in the “mediocre” range, including “Improved sleep”. Still, the authors conclude that “most of the participants were very satisfied with the treatment and the app”. I think it would be important to point out the aspects on which the participants were less satisfied. The conclusion should also be adjusted accordingly to this somewhat low subjective global impression of sleep improvements.

7. Please identify in the text on which instruments the post-treatment values were still in the clinical range.

Discussion

8. In the study limitations, when discussing the other factors that could explain improvements over time, which could not be controlled for, given the lack of a control group, I would add the effect of treatment expectancies.

9. The presence of clinically meaningful sleep problems was not required for participation. Also, the authors recognize that insomnia and sleep difficulties were not severe at baseline. In my first review, I asked the authors to take that into account when interpreting the data. Indeed, one may not expect the same magnitude of effect sizes if a significant proportion of participants were non-clinical cases at baseline (and if insomnia scores were mild). On this, the authors replied that it probably did not make any difference. Yet, effect sizes obtained appear smaller to me than what is generally found in CBT-I trials. What would be the other reasons for that? For me, the most likely explanation remains the low insomnia severity at baseline but if the authors still disagree then I would expect them to identify other reasons.

10. Please tone down the conclusion (last paragraph) and emphasize the preliminary nature of the findings (as in the abstract).

Reviewer #3: The Authors have addressed nearly all question raised. I believe that one issue, however, still needs to be further addressed. Again I specify that the psqi does not define sleep disorders; even if the authors have elaborated on this issue, it is incorrect to speak of scores >5 as sleep disorders; I believe that it should be interpreted as sleep difficulties, at most.

Furthermore it should be specified on what bases sample is divided in sleep disturbances and chronic sleep problems; I suppose that sleep disturbante is PSQI >5, buy chronic sleep problems?

**Do you want your identity to be public for this peer review?** For information about this choice, including consent withdrawal, please see our Privacy Policy

Reviewer #2: No

Reviewer #3: No

---

## [Decision Letter · Decision Letter 2]

9 Apr 2025

Dear Dr. Kurapov,

We look forward to receiving your revised manuscript.

Kind regards,

Serena Scarpelli

Academic Editor

PLOS ONE

Journal Requirements:

Reviewers' comments:

Reviewer's Responses to Questions

**Comments to the Author**

Reviewer #1: (No Response)

Reviewer #2: (No Response)

2. Is the manuscript technically sound, and do the data support the conclusions?

Reviewer #1: (No Response)

Reviewer #2: Yes

3. Has the statistical analysis been performed appropriately and rigorously?

Reviewer #1: (No Response)

Reviewer #2: Yes

4. Have the authors made all data underlying the findings in their manuscript fully available?

Reviewer #1: (No Response)

Reviewer #2: (No Response)

5. Is the manuscript presented in an intelligible fashion and written in standard English?

Reviewer #1: (No Response)

Reviewer #2: Yes

Reviewer #1: I think that in this round of revision the manuscript implemented its quality.

However, there are still some issues I recommend.

Regarding the “sleep disturbance” issues, now it is clearer. However, I suggest replacing “sleep disturbances” with “sleep problems” or “sleep difficulties”, throughout the text. With this terms there is no confusion or ambiguity with “sleep disorders”.

Line 34: I suggest removing “at least”. The impact of stressful situations and stressors in general imply- besides the presence of objective event- a subjective evaluation of the experience, that is less “tremendous scale” occurrence could also result in stress response involving sleep and health issues.

Line 43: the full stop at the end of sentence is missing.

Line 88: please insert reference.

Line 92: maybe “low intensity” is not the right term, please change it and explain better.

Line 167: Fig. 1 description should be below the figure, not above.

Line 315: I suggest inserting data of t0 also in the text, besides the tables, regarding the baseline symptoms.

Line 356: specify also in the text the values of improvements of symptoms.

Line 362: indicate the percentual of subjects that remained at clinical cutoff levels.

Line 393: please report the values of subjects that experienced discomfort in sensor use.

Line 401: it is not correct to say that “veridical feedback on objective sleep parameter can be very helpful for insomnia” since insomnia is a subjective disorder and does not require polysomnography for evaluation or treatment; it is recommended only in case is suspected other sleep disorder in differential diagnosis.

Line 419-421: are not available nap info from the diaries?

Line 435: also this 9% of subjects completed the program?

Line 437: specify the “elevated levels” values of sleep difficulties (i.e., the PSQI score), also for chronic sleep problems.

455: please specify the different level of insomnia with the study referenced (42).

Line 465-467: please delve a little deeper by explaining the subjective nature of insomnia disorder.

Reviewer #2: The authors have successfully incorporated the suggested modifications, providing additional explanations where necessary and refining the presentation of their findings. The revised version now effectively addresses the concerns raised in prior reviews, making the manuscript stronger and more compelling.

**Do you want your identity to be public for this peer review?** For information about this choice, including consent withdrawal, please see our Privacy Policy

Reviewer #1: No

Reviewer #2: No

---

## [Author Response · Author response to Decision Letter 3]

17 Apr 2025

We highly appreciate the feedback, all sections of the manuscript have been appropriately revised as suggested. Please see attached file "Response to Reviewers" where we provide all necessary details regarding every comment. Hope we have covered all concerns this time.

---

## [Editor Report · Decision Letter 3]

24 Apr 2025

Non-guided, Mobile, CBT-I-based Sleep Intervention in War-torn Ukraine: A Feasibility Study

PONE-D-24-35475R3

Dear Dr. Kurapov,

We’re pleased to inform you that your manuscript has been judged scientifically suitable for publication and will be formally accepted for publication once it meets all outstanding technical requirements.

Kind regards,

Serena Scarpelli

Academic Editor

PLOS ONE
---

## [Editor Report · Acceptance letter]

PONE-D-24-35475R3

PLOS ONE

Dear Dr. Kurapov,

I'm pleased to inform you that your manuscript has been deemed suitable for publication in PLOS ONE. Congratulations! Your manuscript is now being handed over to our production team.

Kind regards,

on behalf of

Dr. Serena Scarpelli

Academic Editor

PLOS ONE